# TMT-Based Quantitative Proteomics Analysis Reveals Differentially Expressed Proteins between Different Sources of hMSCs

**DOI:** 10.3390/ijms241713544

**Published:** 2023-08-31

**Authors:** Marie Naudot, Julie Le Ber, Paulo Marcelo

**Affiliations:** 1UR7516, CHirurgie, IMagerie et REgénération Tissulaire de l’Extrémité Céphalique (CHIMERE), Université de Picardie Jules Verne, 80039 Amiens, France; marie.naudot@u-picardie.fr; 2PLATANN, Université de Picardie Jules Verne, 80039 Amiens, France; julie.le.ber@u-picardie.fr; 3Plateforme d’Ingénierie Cellulaire & Analyses des Protéines ICAP, FR CNRS 3085 ICP, Université de Picardie Jules Verne, 80039 Amiens, France

**Keywords:** mesenchymal stem cells, proteomics, tissue engineering, regenerative medicine

## Abstract

Mesenchymal stem cells (MSCs) are an attractive therapeutic tool for tissue engineering and regenerative medicine owing to their regenerative and trophic properties. The best-known and most widely used are bone marrow MSCs, which are currently being harvested and developed from a wide range of adult and perinatal tissues. MSCs from different sources are believed to have different secretion potentials and production, which may influence their therapeutic effects. To confirm this, we performed a quantitative proteomic analysis based on the TMT technique of MSCs from three different sources: Wharton’s jelly (WJ), dental pulp (DP), and bone marrow (BM). Our analysis focused on MSC biological properties of interest for tissue engineering. We identified a total of 611 differentially expressed human proteins. WJ-MSCs showed the greatest variation compared with the other sources. WJ produced more extracellular matrix (ECM) proteins and ECM-affiliated proteins and proteins related to the inflammatory and immune response processes. BM-MSCs expressed more proteins involved in osteogenic, adipogenic, neuronal, or muscular differentiation and proteins involved in paracrine communication. Compared to the other sources, DP-MSCs overexpressed proteins involved in the exocytosis process. The results obtained confirm the existence of differences between WJ, DP, and BM-MSCs and the need to select the MSC origin according to the therapeutic objective sought.

## 1. Introduction

MSCs were initially isolated from bone marrow (BM) based on their ability to adhere to plastic culture dishes and to form colony-forming unit fibroblasts (CFU-Fs) [1]. They are capable of self-renewal, produce an extracellular matrix (ECM), and are able to differentiate into multiple cell types [2]. BM-MSCs have been shown to have immunomodulatory properties, controlling inflammation and modifying nearby immune cells [3,4]. All of these properties have made MSCs a prime candidate for tissue engineering and regenerative medicine. BM-MSC transplants have, therefore, been shown to be beneficial for cartilage regeneration [5], bone tissue regeneration [6], and in acute and chronic models of muscle degeneration [7].

Tissue engineering aims to replace, restore, maintain, or improve the function of human tissues through the laboratory production of biological substitutes for transplantation. In order to create these biological substitutes, the cell part is composed of stem cells, which are used primarily for their ability to differentiate into a desired cell type. MSCs, for example, are widely used for their ability to differentiate into osteoblasts to generate a bone substitute [8]. In this domain, an approach consisting of using only one MSC capacity (e.g., BMP2 synthesis, osteo-differentiation) has provided only limited results [9,10] for good reason: mimicking the in vivo environment (the aim of tissue engineering) means reproducing a complex micro-environment. MSCs are good candidates since they produce an ECM, which participates in the establishment and maintenance of tissues and secrete many factors that promote, among other things, homing [11].

Since the use of bone marrow-derived MSCs, it is recognized that MSC populations can be isolated from a variety of tissues, including adipose tissue, muscle, tendon, peripheral blood, umbilical cord, skin, dental tissue, etc. [12,13,14,15]. MSCs isolated from umbilical cords (UC-MSC), adipose tissue (AT-MSC), or dental pulp (DP-MSC) have significant advantages over BM-MSCs, such as a painless collection, easy extraction, or a high proliferation capacity. For these reasons, these MSCs now tend to replace BM-MSCs in tissue engineering and are beginning to be used in regenerative medicine [16,17,18,19]. For example, (i) in regenerative medicine, osteonecrosis of the femoral head was treated with intra-arterial infusion of UC-MSCs [16], (ii) in tissue engineering, adipose-derived stem cells (ASCs) were combined with 3D porous sponge matrices to improve wound repair [17], and (iii) in cell therapy, the J-REPAIR study: a randomized, double-blind, placebo-controlled study evaluated the efficacy and safety of JTR-161 (an allogenic cell-based product consisting of human dental pulp stem cells) in patients with acute ischemic stroke [19].

But do MSCs from different tissues have exactly the same capabilities? Studies first compared different sources of MSCs (mainly UC-MSC, AT-MSC, and BM-MSC) on their ability to form colonies (CFU-F), their expansion potential, their differentiation capacity, and their cell surface marker expressions [20,21,22,23,24]. Apart from their proliferative potential, no major differences were found between these different MSC sources. Other studies then took the analysis further using omics-based methods: comparisons of the transcriptome, proteome, secretome, and even exosomes of these different MSC sources were performed [25,26,27,28,29]. Indeed, the expression of genes and proteins is known to have an important role in tissue specificity. BMSCs, ASCs, and human umbilical cord perivascular cells differed in their secretion of neurotrophic, neurogenic, axon guidance, axon growth, and neurodifferentiative proteins, as well as proteins with neuroprotective actions against oxidative stress, apoptosis, and excitotoxicity [25]. Some studies have shown that BMSCs, ASCs, and human umbilical cord perivascular cells differed in their secretion of neurotrophic, neurogenic, axon guidance, axon growth, and neurodifferentiative proteins, as well as proteins with neuroprotective actions against oxidative stress, apoptosis, and excitotoxicity [25]. Moreover, bone marrow MSC-derived exosomes had superior regeneration ability, and adipose tissue MSC-derived exosomes played a significant role in immune regulation, whereas umbilical cord MSC-derived exosomes were more prominent in tissue damage repair [27]. Furthermore, comparative transcriptome–proteome analyses of UC-MSCs, AD-MSCs, and BM-MSCs revealed that UC-MSCs promote a more robust host innate immune response; in contrast, adult MSCs appeared to facilitate remodeling of the ECM with stronger activation of angiogenic cascades [29].

In the present study, we made a proteomic comparison of three different sources of MSCs using the TMT-based quantitative technique. Our analysis focused on the MSC biological properties of interest for tissue engineering and regenerative medicine. Indeed, MSCs used in innovative therapy and mainly in tissue engineering must be able to recreate (alone or by the paracrine effect) the injured tissue: (i) by differentiating into cells of interest, (ii) by producing an ECM, and/or (iii) by homing thanks to secreted proteins and finally, (iv) they must limit inflammation and modulate the immune response.

We compared three sources of MSCs: umbilical cord-derived Wharton’s jelly MSCs (WJ-MSC), dental pulp MSCs (DP-MSCs), and bone marrow MSCs (BM-MSCs), cultivated under proliferative conditions. The objective was to identify whether there are sufficient proteomic differences to make an MSC source more attractive for a given cell therapy application.

## 2. Results

The TMT-based quantitative proteomics technique we used enabled us to analyze the proteins present in all of the samples.

### 2.1. Qualitative Analyses of the Three Sources of hMSCs

#### 2.1.1. Differentially Expressed Proteins and Biological Processes Involved

We identified a total of 611 differentially expressed human proteins. The biological processes affected by these variations of expression between cell sources are shown in Figure 1. The biological process with the greatest variation in protein expression (outside of the “other metabolic or biological” groups) was cellular organization and biogenesis at 12%.

#### 2.1.2. Group Comparison

Pairwise comparison: BM vs. DP, BM vs. WJ, and DP vs. WJ (Figure 2) showed a stronger difference in protein expression (either up- and down-expressed) between BM-MSCs and WJ-MSCs (Figure 2a). In contrast, BM vs. DP had the fewest variant proteins (Appendix A). These results can be explained by the fact that BM-MSCs and DP-MSCs are both adult MSCs.

The GO classification of differentially expressed proteins with comparisons: BM vs. DP, BM vs. WJ, and DP vs. WJ (Figure 2b) highlight several features: comparisons of BM vs. DP and BM vs. WJ revealed many underexpressed proteins in the majority of biological processes. This suggests that BM-MSCs expressed fewer proteins involved in different biological processes than other cell sources, with the exception of signal transduction. The pairwise comparison of DP vs. WJ showed overexpression of proteins involved in nine of the thirteen biological processes listed In Figure 2b. DP-MSCs overexpress more proteins than other sources for the following biological processes: the cell cycle or cell proliferation, cell organization and biogenesis, developmental processes, stress response, and transport and DNA metabolism. WJ-MSCs overexpressed proteins in the major processes: cell adhesion, protein metabolism, RNA metabolism and transcription, and cell–cell signaling (Appendix A).

### 2.2. Analyses Focused on the Functions of Interest of MSCs in Cell Therapy

#### 2.2.1. MSC Characteristics

The International Society for Cellular Therapy has defined MSCs as cells with a specific immunophenotype, ex vivo plastic-adherent growth, and multilineage differentiation [30]. We, therefore, looked for differences in protein expression from these features.

CD markers of MSCs

Among the CD marker profiles, four membrane proteins had differences in expression between MSC sources, with three in the CAM family (Table 1). BM-MSCs expressed the most Endoglin: CD105, VCAM1: CD106 and MCAM: CD146 compared with other sources. WJ-MSC down-expressed VCAM1: CD106 and ALCAM: CD166 compared with other sources. More than just cell markers, these proteins are involved in important processes, such as angiogenesis and immune response.

Differentiation capacities

Some proteins have been identified in those differentially expressed between the different sources of MSCs as being involved in osteoblastic, adipocyte, neuronal, or muscle differentiation (Table 2). Alkaline phosphatase, an important enzyme in bone mineralization, was underexpressed by WJ-MSCs compared with DP-MSCs and BM-MSCs. Compared with the others, WJ-MSCs produced more protaglandin G/H synthase 2, while BM-MSCs produced more of the adipogenesis regulatory factor. BM-MSCs seemed to be the most capable of ensuring adipocyte differentiation as opposed to WJ-MSCs. Regarding other types of differentiation, it seems that BM-MSCs, which overexpressed more proteins, were more capable of neuronal or muscle differentiation.

#### 2.2.2. ECM Production

The ECM is a major component of the cellular micro-environment. It is composed of structural components (collagens, ECM glycoproteins, and proteoglycans) and ECM-associated proteins. The production of the ECM by MSCs is one of the criteria for selecting them for tissue engineering.

Collagens, ECM glycoproteins, and proteoglycans

The ECM structural components differentially expressed in the pairwise comparisons of BM/DP, BM/WJ, and DP/WJ, as shown in Figure 3.

Collagen is the most abundant fibrous protein in the ECM. There are 28 different types in humans, and in our study, 11 collagens were identified as being differentially expressed in the three MSC sources. Col14A1 was most highly expressed by BM-MSC. WJ-MSC most expressed Col3A1, Col4A1, Col4a2, Col5A1, and Col16A1. DP-MSCs underexpressed Col4A1, Col8A1, and Col11A1 compared with other sources.

In our study, among the ECM glycoproteins identified as the statistically variant members of the insulin-like growth factor binding protein (IGFBP) family, IGFBP-4, IGFBP-5, and IGFBP-7 were found to be highly expressed by BM-MSCs compared with DP-MSCs. BM-MSCs overexpressed the MFGE8 protein compared with the other sources of MSCs. DP-MSCs strongly expressed fibrilin-2, known to regulate the early process of elastic fiber assembly. This protein also regulates osteoblast maturation by controlling TGF-beta bioavailability and calibrating TGF-beta and BMP levels. WJ-MSC overexpressed fibronectin, EMILIN-1, fibulin-2, IGFBP7, and nidogen-2 compared with the other sources.

For proteoglycans, our analysis revealed a strong variation in the expression of a number of proteoglycans in WJ-MSCs. WJ-MSCs strongly underexpressed three of the four members of the Syndecan family. Conversely, they overexpressed the other identified proteoglycans: Biglycan, Tenascin, and VCAN. DP-MSCs overexpressed Syndecan-1 and Syndecan-4 compared with BM- and WJ-MSCs.

ECM-associated proteins

All differentially expressed proteins known to be associated with, interact with, or regulate the ECM are listed in Figure 4 (and Appendix A).

Most of the ECM regulators presented here are enzymes or proteins involved in ECM remodeling through bond formation or degradation. In the BM/DP comparison, among the nine proteins differentially expressed, all proteins are overexpressed by BM-MSCs, with the exception of neprilysin. DP-MSCs were found to overexpress just one protein compared with the other two sources: neprilysin, a protein involved in elastin degradation. WJ-MSCs overexpressed a number of proteins compared with other sources, such as ADAMTSL, dipeptidyl peptidase 4, ITIH1, ITIH2, SERPINE1, or TGM2.

The group of ECM-affiliated proteins was largely made up of the integrin family. The expression of the various integrins changed from source to source of the MSCs, with no particular trend. WJ underexpressed integrin beta-like protein 1, a protein that promotes cell migration.

#### 2.2.3. Cell–Cell Signaling

Proteins belonging to the biological process “cell-cell signaling” (GO term) identified as a significant variant between the different MSC sources are shown in Figure 5.

The ratios with WJ-MSCs showed the most significant differences in protein expression. Compared with BM and DP, WJ-MSCs overexpressed CXCL6, GDF15, GPC6, HMGA2, INHBA, NAMPT, PLAT, and PTGS2, and for some of them, very strongly. WJ-MSCs underexpressed only three proteins: CAV1, DAB2, and PLPP3, compared with other cell sources. DP-MSCs overexpressed three proteins: AMPH, NXN, and SDC1, and underexpressed JUP. BM-MSCs overexpressed GPC4 but underexpressed more proteins: CTHRC1, GREM1, and LTBP4 (Appendix A).

#### 2.2.4. Inflammation and Immune Response

MSCs have been shown to secrete a broad spectrum of bioactive molecules that induce a variety of responses, including the inhibition of inflammatory and/or immune responses. We identified seven proteins involved in the inflammatory response and eleven proteins in the immune response (Figure 6).

BM-MSCs overexpressed PTGES3 when DP-MSCs overexpressed follistatin and galectin-3, compared with the others. WJ-MSCs overexpressed PTGS2 and PTX3 compared with BM and DP-MSCs. The differences between cell sources were most marked in the immune response with the presence of important proteins such as ADGRE5, ICAM1, CD200, VCAM1, or DPP4. WJ-MSCs overexpressed ICAM1, CD200, CXCL6, and DPP4 compared with BM and DP-MSCs but underexpressed ALCAM, CAMK1D, VCAM1, and SERPINB1. BM-MSCs only overexpressed VCAM1 compared with the two others. DP-MSCs overexpressed Nectin-3 and underexpressed CD200 and DDP4 compared with BM and WJ-MSCs (Appendix A).

## 3. Discussion

TMT-based quantitative proteomic analysis was performed on MSCs grown under conventional culture conditions. We did not generate any particular condition, such as hypoxia treatment, that could generate a response to the stimuli. The aim of this study was to highlight differences in level protein expression and only the proteins produced by all of the samples were analyzed.

The comparison of our study with other articles is not obvious since no published study to date has performed the same proteomic analysis of BM-MSCs, WJ-MSCs, and DP-MSCs. One team compared human mesenchymal stem cells derived from dental pulp, bone marrow, adipose tissue, and umbilical cord tissue by the expression of 15 pluripotent stem cell genes [23]. There are articles on comparative proteomic analysis, but the cell sources are not the same. Our comparison with the literature can only be partial. Unfortunately, we were unable to include AT-MSCs in this study because we were unable to obtain adipose tissue.

Qualitative analyses of the three MSC sources revealed a number of differentially expressed proteins of the same order of magnitude as those found in other articles [27,28]. Comparing these articles, biological processes impacted by these differences in expression are not always the same (it depends in part on the cell sources), but we have in common among the most variant: “cell organization and biogenesis” and “secretion by cell”, which for us was included in the group “other biological process”.

In the group comparison, the Venn diagram shows a greater difference between WJ-MSCs and other sources. This result can be explained by the fact that BM and DP are both adult MSCs, whereas WJ-MSCs have a fetal origin. Shin et al. showed that the secretome of fetal-derived MSCs, such as PL and WJ, had a more diverse composition than that of AD- and BM-derived MSCs [28].

In analyzing the quantitative results, we chose to focus on the elements that make MSCs a tool of choice for cell therapy.

CD markers are classically analyzed by cytometry. This type of analysis highlights the positivity or otherwise of these cells to the CD markers sought. Proteomic analysis gives a much more precise idea of the expression of these markers. We were, therefore, able to show that BMs are the source of cells that express the most CD105, CD106, and CD146, while WJs express CD106 and CD166 the least. This difference in expression could reflect a preference for interaction. The function of these markers on the MSC surface is still poorly understood. We hypothesize that the difference in expression observed in BM-MSCs compared with other sources could be because they belong to the hematopoietic niches [31].

Another important feature of MSCs is their ability to differentiate into various cellular types. This is the point most studied during MSC source comparisons. It is important to couple differentiation capacity with proliferation capacity. Indeed, it is known that there is a balance between proliferation and differentiation. Thus, WJ-MSCs, which proliferate faster than BM-MSCs, appear to have little or no capacity to differentiate into osteoblasts or adipocytes [21,22,32]. This is also what we found in our study with WJ-MSCs underexpressing ALPL, a protein that is essential for osteoblastic mineralization, and overexpressing PTGS2, a protein that suppresses adipocyte differentiation. Our results suggested that BM-MSCs are superior to other sources in osteogenic, adipogenic, neuronal, or muscular differentiation. By comparing gene expression profiles, Hsieh and his team found that BM-MSCs were more capable of osteogenic and adipogenic differentiation, while WJ-MSCs proliferated more [33]. In a study comparing fetal MSC sources (WJ, fetal, and the maternal side of a placenta) and adult MSCs (BM, AT), BM-MSCs showed the greatest capacity for differentiation [34]. However, not all articles go in the same direction. For example, Donders and his team showed that WJ overexpressed genes involved in differentiation, maturation, and neuronal support compared with BM-MSCs [35].

In tissue engineering, one of the most important points is the ability of the cells used to generate an extracellular matrix. In addition to providing physical support to cells, the ECM actively participates in the establishment and maintenance of differentiated tissues and organs by regulating growth factors, hydration levels, and the pH of the local environment [36]. Among the ECMs, collagens are well represented in our study. WJ-MSCs overexpressed the most collagen types compared to other sources and collagens of different classes: fibrillar, fibrillar-associated collagen with interrupted triple helices, the basement membrane, filamentous, and short chain and multiplexins. Compared with the other two groups, BM-MSCs overexpressed only Col14A1, which is often present in areas of high mechanical stress, indicating that it potentially has a role in maintaining mechanical tissues.

Some ECM glycoproteins, such as fibronectin, are ubiquitous, while others have more specific localizations, such as the laminins of basal membranes. They contain several structural and functional domains, several sites of cell attachment via integrins or other receptors, the most frequent of which contain the Arg-Gly-Asp (RGD) sequence, and several sites of interaction with other extracellular macromolecules. Since they are capable of numerous interactions with the micro-environment, a strong variation in their expression can have a real impact on MSC capacities.

In the glycoproteins identified as differentially expressed, again, it is WJ-MSCs that overexpress the most proteins. However, we noted the presence of three members of the IGFBP family. These proteins are more highly expressed by BM-MSCs than by DP-MSCs. This IGFBP protein family serves as a transport protein for insulin-like growth factor-1 (IGF-1), influences the bioavailability of IGFs, and, therefore, reduces their signaling with cell receptors. IGFs stimulate the proliferation of differentiated chondrocytes, leading to the enlargement of the conjugation cartilage and elongation of the bone. In addition, IGFs promote the growth of all tissues, stimulate protein synthesis, and enhance Ca2+ uptake. IGF-1, therefore, enables growth or at least limits age-related bone loss [37].

Proteoglycans make up the interstitial matter. They form a hydrophilic gel with a wide range of functions, such as tissue hydration, the modulation of signaling pathways, and resistance to tensile forces. In this category, only six proteins have been identified as variants, and more especially by the WJ-MSC. If we look at all of the components of all the ECMs, we can see that WJ-MSCs overexpress most of the players in the basal lamina, such as collagen 4, laminins, nidogens, and integrins. The basal lamina enables epithelial cells to adhere to the underlying connective tissue and constitutes a major interface between epithelial tissue cells and the body’s interior for the regulation and diffusion of nutrients. It also plays a role in the survival, proliferation, and differentiation of cells in the various epithelial tissues [36].

Among the ECM regulators, BM and WJ-MSCs shared the overexpression of the various proteins identified. ECM-affiliated proteins were mainly represented by the integrin family. To be functional, integrins must form heterodimers (composed of an alpha and beta chain). Here, no functional heterodimer appears to be over- or underexpressed. In fact, the alpha integrins were identified as dimerized with the beta 1 integrin, which showed no difference in expression. Similarly, integrin beta 3 normally associates with alphaIIb or alpha V, which were not present in our analysis. Apart from the integrins, we can see the overexpression of galectin-3 by the DP-MSC compared with BM and WJ. Galectin-3 modulates important interactions between epithelial cells and the extracellular matrix, which promotes tissue vascularization [38].

Proteins belonging to the biological process “cell-cell signaling” are largely composed of transcriptional regulators, growth factors, or hormone regulators. A number of Wnt pathway players are presented: FZD7, NXN, RECK, and WNT5A. The Wnt signaling pathway is very important in MSCs (as Notch). It is involved in osteoblastic, adipocytic, and chondrocytic differentiation [39,40]. Above all, it plays a role in MSC tissue regeneration [41].

Two proteins involved in exocytosis are also differentially expressed: SDC1 and AMPH. They are overexpressed by DP-MSCs compared with BM and WJ. Exosomes secreted by MSCs have been the subject of recent studies. They are proving to be one of the main mechanisms of the therapeutic action of MSCs, which has so far been neglected [42,43,44]. Comparative proteomic analysis of exosomes from three MSC sources revealed differences in capacity. This would justify the choice of an MSC source based on potential applications [27].

Our study did not reveal differences in the expression of important factors secreted by MSCs, such as the proteins involved in angiogenesis, HGF, IGF-1, MCP-1, angiogenin, or VEGF, or the proteins involved in hematopoiesis, TGFB1, TGFB2, GDF6, VEGF-C, M-CSF, CSF, or interleukins.

In contrast, expression differences were observed for proteins involved in inflammatory and/or immune responses. Within these two categories, WJ-MSCs showed the greatest variation in protein expression. Like Donders et al., we found the overexpression of CD200, ICAM-1 by WJ-MSCs [35]. Although the main secreted factors responsible for the immunological and anti-inflammatory competence of MSCs described in the literature did not emerge in our study [45,46], the proteins identified here suggest that WJ-MSCs are among the three sources tested and are the most likely to have an action on the inflammatory and immune response.

## 4. Materials and Methods

### 4.1. Cell Cultures

Human cells used in this study come from cryotubes made after isolation and expansion of MSCs from bone marrow, dental pulp, and umbilical cords.

Five different donors by source were used in the present study. For all sources (BM-MSCs, DP-MSCs, WJ-MSCs), cell expansion was performed in a modified Eagle’s medium (Sigma-Aldrich, Saint-Quentin Fallavier, France) supplemented with 10% fetal bovine serum, 2-mM L-glutamine, and 1% antibiotics (100 U/mL penicillin and 100 pg/mL streptomycin). All supplements were purchased from Eurobio (Courtaboeuf, France). For the BM-MSC culture, a basic fibroblast growth factor (bFGF, TebuBio, Le Perray-en-Yvelines, France) was added to the supplemented medium at 2 ng/mL. Cells (at passage 3 or 4 for WJ and DP and passage 2 or 3 for BM) were seeded at a density of 1.10^6^ cells in T175 flasks. They were cultured in a humidified 5% carbon dioxide (CO_2_) atmosphere at 37 °C. The medium was refreshed once a week. By MSC sources and donors, once 80% confluence had been reached (average 7 days for WJ-MSCs and DP-MSCs, 2–3 weeks for BM-MSCs), the culture medium was removed, and cells were rinsed with 10 mL of phosphate-buffered saline (PBS, Corning, Bagneaux sur Loing, France). One T175 flask was used for cell characterization, and the other for protein extraction. Cells used for characterization were suspended after 5 min of trypsin action (trypsin 0.25% + EDTA 0.02%, Pan Biotech, Bernolsheim, France), followed by a centrifugation step. Cells dedicated to protein extraction were directly recovered by a cell scraper in 5 mL of PBS.

### 4.2. MSCs Characterization

Cultured cells were harvested, and flow cytometry was performed for phenotypic characterization. Briefly, MSCs were suspended in PBS and characterized by using the following antibodies: CD90-FITC, CD105-PE, CD106-PE, CD146-FITC, CD166-PE, CD45-APC (BD Biosciences, Le Pont-de-Claix, France), and CD73-APC, CD19-FITC, CD34-PE (Miltenyi Biotec, Paris, France). Cells incubated with phycoerythrin (PE)-conjugated (Miltenyi Biotec, Paris, France), fluorescein isothiocyanate (FITC)-conjugated, or allophycocyanin (APC)-conjugated (BD Biosciences) mouse IgG1 isotype antibodies were used as a negative control. Analyses were performed on a MacsQuant Analyser 10 (Miltenyi Biotec, Paris, France), and the results were analyzed using FlowJo software (FlowJo V10, Tree Star, Ashland, OR, USA). Phenotypic identification of cultured BM-MSCs, DP-MSCs, and WJ-MSCs are shown in Appendix A.

### 4.3. Proteins Extraction and TMT Method

Proteins were extracted using an EasyPep sample preparation kit (Pierce, San Jose, CA, USA) using manufacturer recommendations. The proteins were quantified using the Micro BCA kit method (Pierce, San Jose, CA, USA).

Twenty-five micrograms of proteins of each sample were digested and labeled with TMTpro™ 16-plex reagents (Thermo Fisher Scientific, San Jose, CA, USA), mixed in equimolar amounts. A fractionation was purchased using a High pH Reversed-Phase Peptide Fractionation Kit (Pierce, San Jose, CA, USA), according to manufacturer recommendations. The tryptic peptide solutions were dried under vacuum and reconstituted in 20 µL water/1% formic acid (*v*/*v*) each.

### 4.4. LC-MS/MS and Data Analyis

The LC–MS/MS platform consisted of an Ultimate 3000 RSLC UPLC system coupled with an Orbitrap Fusion mass spectrometer (MS) (ThermoFisher Scientific, San Jose, CA, USA) with a nano-trap column (Acclaim PepMap 100 Å C18, 5 µm, 100 µm i.d. × 2 cm length, ThermoFisher Scientific) and an Easy-Spray column (Acclaim PepMap 100 Å C18, 2 µm, 75 µm i.d. × 50 cm length, ThermoFisher Scientific). Ten fractions of the TMT-labeled digest were separated by on-line nanoLC and analyzed by nano-electrospray tandem mass spectrometry. The overall workflow of the analysis is presented in Figure 1. The peptide mixtures were injected onto a nano-trap column with a flow of 5 µL/min and subsequently gradient-eluted with a flow of 300 nL/min, from 4% to 30% acetonitrile (*v*/*v*) for 140 min. Each fraction was analyzed on an Orbitrap Fusion MS using synchronous precursor selection (SPS) MS3 quantitation. The full scan was performed in the range of 375–2000 *m*/*z* at a nominal resolution of 120,000 at 200 *m*/*z* and AGC set to 4.105, followed by the selection of the most intense ions above an intensity threshold of 5000 for collision-induced dissociation (CID)-MS2 fragmentation in the linear ion trap with 35% normalized collision energy. The isolation width for the frontal cortex samples was set to 0.7 *m*/*z* with no offset. The top 10 fragment ions for each peptide MS2 were notched out with an isolation width of 2 *m*/*z* and co-fragmented to produce MS3 scans analyzed in the MS at a nominal resolution of 50,000 after higher-energy collision dissociation (HCD) fragmentation at a normalized collision energy of 65%. Data were processed using Proteome Discoverer 2.5 (ThermoFisher Scientific, Bremen, Germany) before being run against the *Homo sapiens* Uniprot database (release 2022_12). Parameters were specified as follows: trypsin enzyme, two miscleavages allowed, minimum peptide length of six amino acids, TMT tags on lysine residues and peptide N-termini (+304.207 Da), carbamidomethylation of cysteine residues (+57.021 Da) as fixed modifications, oxidation of methionine residues (+15.995 Da) and acetylation of protein N-termini (+42.011 Da) as variable modifications, precursor mass tolerance of 10 ppm, and a fragment mass tolerance of 0.6 Da. Peptide spectral match (PSM) error rates were determined using the target-decoy strategy coupled with percolator modeling of true and false matches [47]. Reporter ions were quantified from MS3 scans using an integration tolerance of 20 ppm with the most confident centroid setting. An MS2 spectral assignment false discovery rate (FDR) of less than 1% was achieved by applying the target-decoy strategy. Following spectral assignment, peptides were assembled into proteins and were further filtered based on the combined probabilities of their constituent peptides to a final FDR of 1%. In addition, we only validated the proteins that were present in the five biological replicates. In cases of redundancy, shared peptides were assigned to the protein sequence with the most matching peptides, thereby adhering to the principles of parsimony. The DAPs were identified based on a *t*-test with a *p*-value less than 0.05 and with a fold change >2 or <0.5 (Appendix A). The mass spectrometry proteomics data have been deposited to the ProteomeXchange Consortium (http://www.proteomexchange.org (accessed on 19 July 2023) via the PRIDE partner repository with dataset identifier PXD043912 [48].

## 5. Conclusions

In this study, we succeeded in highlighting the differences in the proteomic expression of three MSC sources and linking these differences with their applicative interest for regenerative medicine and tissue engineering. WJ-MSCs showed the greatest variation compared with the other sources. WJ produced more ECM proteins or ECM-affiliated proteins and produced more proteins related to the inflammatory and immune response processes. BM-MSCs expressed more proteins involved in osteogenic, adipogenic, neuronal, or muscular differentiation and proteins involved in paracrine communication. Compared to the other sources, DP-MSCs overexpressed proteins involved in exocytosis processes. None of the MSC sources are without interest. The results obtained confirm the need to select the origin of MSCs according to the desired therapeutic objective.

## Figures and Tables

**Figure 1 ijms-24-13544-f001:**
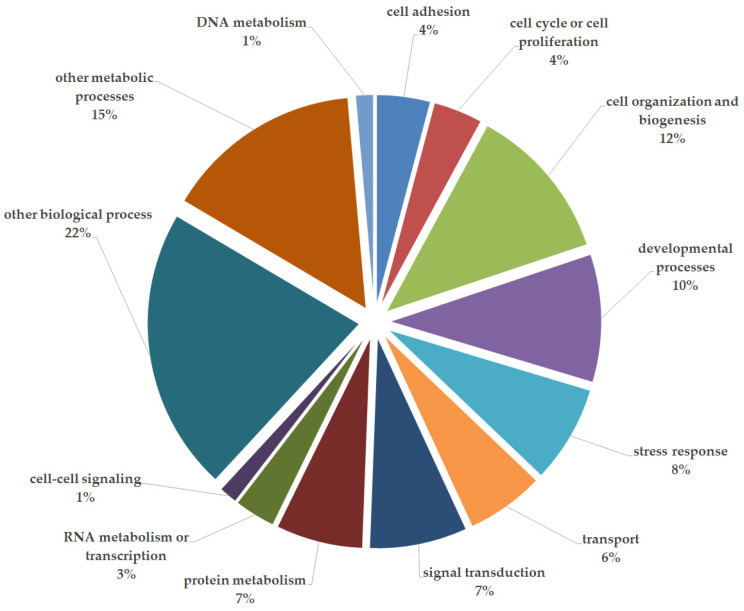
Biological process classification of differentially expressed proteins.

**Figure 2 ijms-24-13544-f002:**
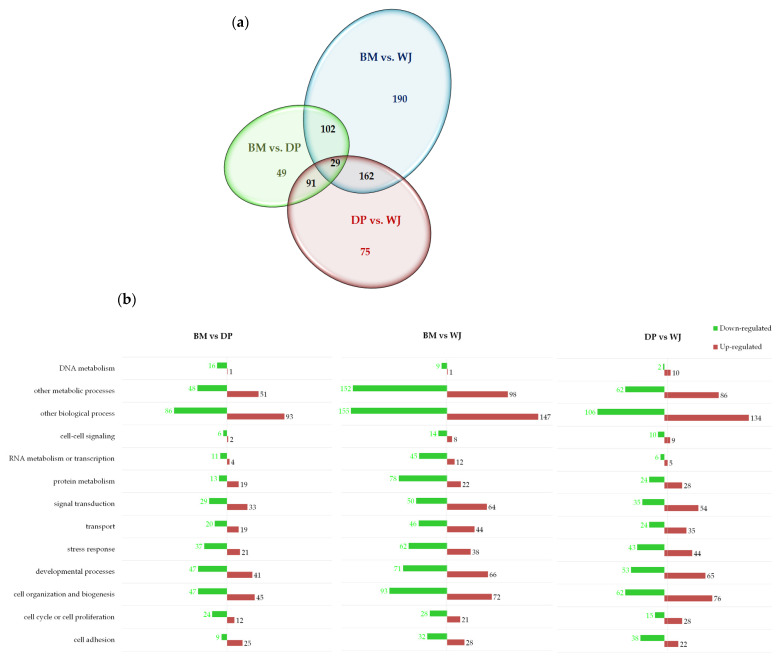
(**a**) Venn diagram of differentially expressed proteins, (**b**): GO classification of differentially expressed proteins using pairwise comparison: BM vs. DP, BM vs. WJ, and DP vs. WJ.

**Figure 3 ijms-24-13544-f003:**
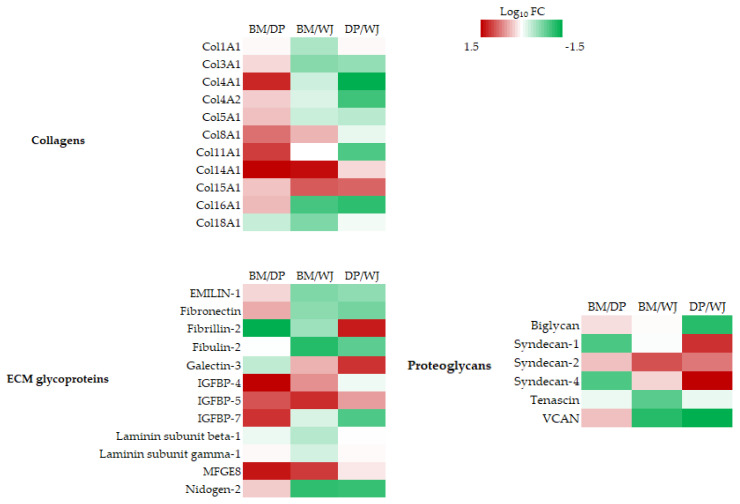
Heatmaps of differentially expressed proteins in the following categories: collagens, proteoglycans, or extracellular matrix glycoproteins, using pairwise comparison: BM/DP, BM/WJ, and DP/WJ. Color coding in the heatmap depicts the variation between the maximum (coded in red tones) to the minimum (coded in green tones) observed on Log_10_FC, and statistically insignificant results are color-coded white.

**Figure 4 ijms-24-13544-f004:**
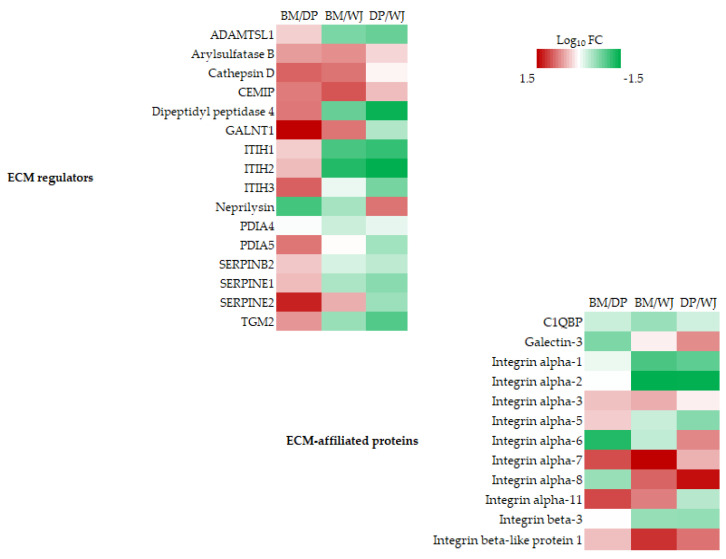
Heatmaps of differentially expressed proteins in the following categories: ECM regulators and ECM-affiliated proteins, using pairwise comparison: BM/DP, BM/WJ, and DP/WJ. Color coding in the heatmap depicts the variation between the maximum (coded in red tones) to the minimum (coded in green tones) observed on Log_10_FC, and statistically insignificant results are color-coded white.

**Figure 5 ijms-24-13544-f005:**
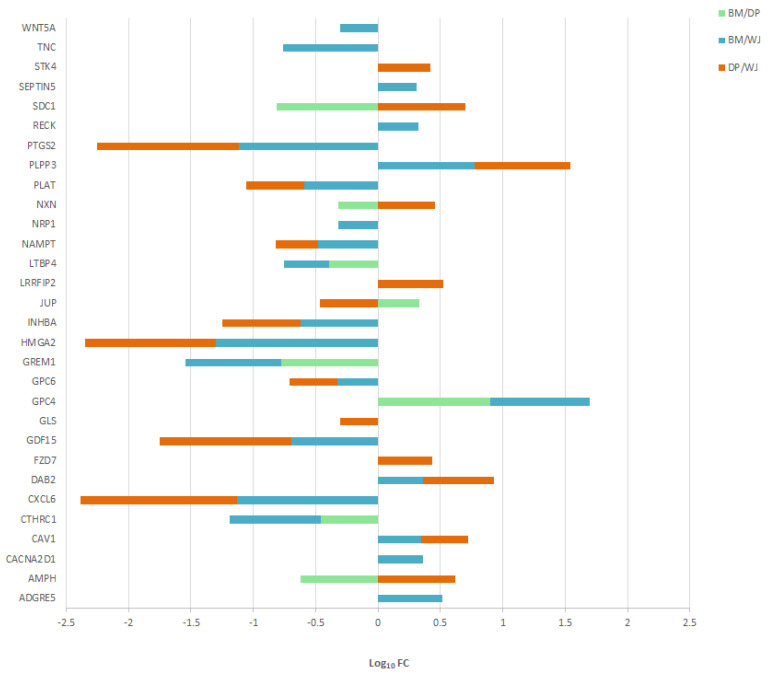
Proteins involved in cell–cell signaling significantly underexpressed (Log_10_FC negative) or overexpressed (Log_10_FC positive) according to BM/DP, BM/WJ, and DP/WJ ratios. Only statistically significant values are shown.

**Figure 6 ijms-24-13544-f006:**
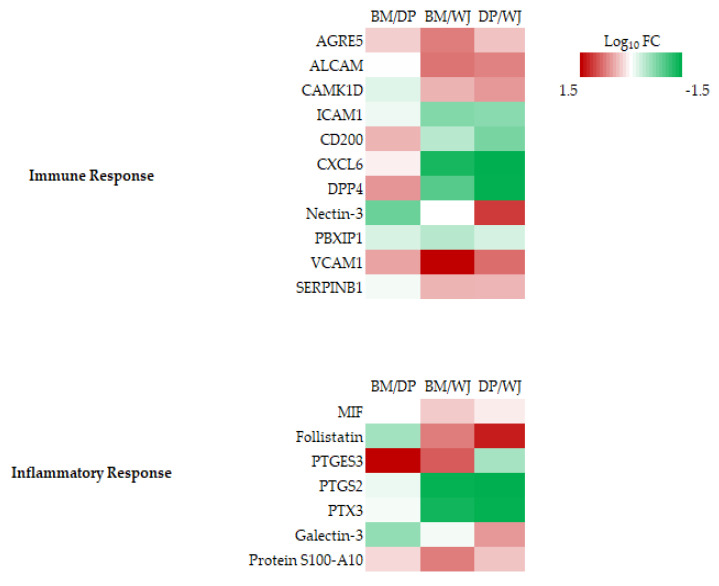
Heatmaps of differentially expressed proteins: inflammatory response and immune response using pairwise comparison: BM/DP, BM/WJ, and DP/WJ. Color coding in the heatmap depicts the variation between the maximum (coded in red tones) to the minimum (coded in green tones) observed on Log_10_FC, and statistically insignificant results are color-coded white.

**Table 1 ijms-24-13544-t001:** CD proteins differentially expressed in MSCs sources. Only statistically significant values are shown.

Protein Name	Ratio			Most Expressed by	Least Expressed by
	BM/DP	BM/WJ	DP/WJ		
CD105: Endoglin	2.842	2.542	-	BM	
CD106: VCAM1	2.419	9.136	3.776	BM	WJ
CD146: MCAM	2.689	2.477	-	BM	
CD166: ALCAM	-	3.567	3.145		WJ

**Table 2 ijms-24-13544-t002:** Differentially expressed proteins in MSCs playing a role in cell differentiation. Only statistically significant values are shown.

Protein Name	Gene Symbol	Ratio BM/DP	Ratio BM/WJ	Ration DP/WJ	Role of Protein
**Osteoblast differentiations**					
Alkaline phosphatase, tissue-nonspecific isozyme	ALPL	-	5.508	4.441	Promotes calcification
Transforming growth factor beta-1 proprotein	TGFB1	-	-	0.452	TGF-β/BMP pathway controls the differentiation of mesenchymal precursor cells
Fibronectin	FN1	-	-	0.206	Marker of osteoblast maturation
**Adipocyte differentiations**					
Prostaglandin G/H synthase 2	PTGS2	-	0.077	0.073	Suppressor of adipocytic differentiation
Adipogenesis regulatory factor	ADIRF	3.25	5.483	-	transcriptional regulator of white adipocyte differentiation
**Neuronal differentiations**					
Glia-derived nexin	SERPINE2	6.058	-	0.269	Promotes neurite extension by inhibiting thrombin
Dihydropyrimidinase-related protein 2	DPYSL2	-	2.706	-	Involved in the regulation of axon formation during neuronal polarization, as well as in axon growth and guidance
Echinoderm microtubule-associated protein-like 1	EML1	-	2.598	4.279	Required for normal proliferation of neuronal progenitor cells
Neuronal growth regulator 1	NEGR1	2.263	-	-	May function as a trans-neural growth-promoting factor in regenerative axon sprouting
**Other differentiations**					
Actin2, aortic smooth muscle	ACTA2	2.068	-	-	Involved in vascular contractility and blood pressure homeostasis
Transgelin	TAGLN	3.061	2.052	-	Ubiquitously expressed in vascular and visceral smooth muscle and is an early marker of smooth muscle differentiation
Caldesmon	CALD1	2.139	2.48	-	Regulated actomyosin interactions in smooth muscle and non-muscle cells involved in Schwann cell migration during peripheral nerve regeneration

## Data Availability

The data presented in supplementary figures in the article are available in the Appendix A.

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
