# Peer review of "TMT-Based Quantitative Proteomics Analysis Reveals Differentially Expressed Proteins between Different Sources of hMSCs"

_ijms, 2023, doi:10.3390/ijms241713544_

Round 1
Reviewer 1 Report
The paper could be more interesting because the method used is basically specific but most of the variations detected are in the field of "other metabolic processes" or "other biological processes" which does not mean anything to the reader. Furthermore, cells express RNA following their culture conditions and the authors do not report the culture conditions of cells in terms of confluence and viability. And finally it doesn't seem to show up any useful result for people interested in cell therapies.
Reviewer 2 Report
Manuscript ” TMT-based quantitative proteomics analysis reveals differentially expressed proteins between different sources of hMSCs” by Naudot M et al.
Naudot M and co-authors compare expression of proteins in mesenchymal stromal cells (MSCs) derived from different sources using quantitative mass-spectrometry. MSCs are immunomodulatory cells that can be used in allogeneic setting and can be easily derived from living donors. The primary cells are relevant for contemporary medicine and it is important to understand the difference in MSCs derived from different sources (organs). Although the topic is relevant, the manuscript in its present form has several significant flaws.
Major issues:
1) Materials and Methods part should be significantly expanded. The authors must describe in details how they cultured each type of cells (media and coating of cell culture plates if any), for how long period (number of doublings or number of passages with seeding densities) and how they prepared the samples for mass-spectrometry. A prerequisite for any scientific paper is to provide enough information for anybody who wants to repeat the study. In page 9 line 237, the authors state that “…MSCs grown under conventional conditions.” There are no conventional conditions for culturing MSCs – there are many different culture media and substrata for that. There is a large body of published data showing that expression of proteins in MSCs depends on culture conditions and changes with time during culturing in vitro.
2) All MSC cultures used in the manuscript must be characterized using the International Society for Cellular Therapy (ISCT) criteria. At least expression of CD105, CD73 and CD90, and lack of expression of CD45, CD34, CD14 or CD11b, CD79alpha or CD19 and HLA-DR surface molecules must be demonstrated using fluorescence activated cell sorting (FACS) analysis. In page 10 lines 260-267, the authors claim that proteomic analysis (mass-spec probably) is superior over FACS, which is not true. The methods are complementary to each other. Mass-spec analysis measures overall expression level of proteins in a population of cells. Same expression level measured by mass-spec can be achieved in a homogeneous population where each cell expresses certain marker and in a heterogeneous population with a subset of cells expressing very high levels of that marker. FACS analysis provides information on distribution of expression in the cell population. Therefore, FACS analysis must be performed to ensure that the analyzed cells represent a homogeneous population of MSCs.
3) It is not clear if the authors used one line of each type of MSCs that was cultured in five plates or five different lines. If the former is right, the authors should significantly tone down their claims and conclusions, because the differences might be simply a result of various genetic backgrounds in those cell lines (or culture conditions, or time in in vitro culture).
Minor issues:
1) The authors overestimate the prognostic power of marker expression. For instance, it is not enough to demonstrate higher expression of some anti-inflammatory and immunomodulatory markers to claim the ability of the corresponding cells to better modulate inflammatory and immune responses. To make such a claim, the authors should perform at least a battery of in vitro functional tests on the cells. Same is true for exosome production and differentiation capacity. If the authors cannot demonstrate the functional tests, they need to tone down the claims in the Abstract and Discussion; and only claim the difference in expression of markers related to the processes.
2) Page 10 lines 271-273. The authors state that “Thus, WJ-MSCs, which proliferate faster than BM-MSCs, appear to have little or no capacity to differentiate into osteoblasts or adipocytes.” If it is true, WJ-MSCs, according to the ISCT criteria, do not comply with the definition of MSCs and should be renamed. This is why it would be important for the authors to try to differentiate their WJ-MSCs in fat, cartilage and bone tissues.
3) Although MSCs are also used in regenerative medicine as a source of cells, the majority of clinical trials utilize their immunomodulatory abilities. A paragraph on immunomodulation by MSCs should be added to the Introduction.
4) The Results contain several sentences describing general properties of certain protein groups for instance ECM glycoproteins (page 7 lines 158-164) and proteoglycans (page 7 lines 173-175). They should be moved to the Discussion.
Minor editing of English language required
Reviewer 3 Report
In my opinion the paper of Naudot et al is very interesting. To better study and understand the role of MSC in regenerative medicine and advanced therapy is really important to go deeper to the comprehension of these cell type. This paper give an overview about this argument. Only one point: could you explain why have you choose to not study MSC derived by Adipose Tissue? Please, write something about this in the discussion.
Reviewer 4 Report
Line 48: Replace "..." with "etc".
Line 51-52: Please elaborate on the use of these MSC types and support with examples situations in which they have been demonstrated useful.
Line 57-60: The authors are suggested to elaborate with examples, why it is helpful to evaluate differences between MSC sources in terms of their proteomic profile. Have there been discrpencies between differential potential vs secretome vs CFU vs self-renewal etc.?
Was there a reason the authors did not evaluate ASCs, considering the fact that they are very ubiquitous?
The materials and methods section should be organized into subsections.
It seemed that dental pulp MSCs did not express some of the cell surface markers, can the authors explain why?
Line 266: The authors mentioned that they expect bmMSCs to express significant CD-surface markers due to their niche. Can the authors elaborate why?
Round 2
Reviewer 1 Report
Maintaining my opinion on cell culture conditions and RNA expression under 10% FBS, a huge container of growth factors that could erase the effect of any other variable, I agree to publish the paper as it is. Readers will be able to see at least now all the variables involved in the experimental approach.
Reviewer 2 Report
The manuscript can be published now.